# Low levels of *Plasmodium falciparum* parasitaemia among patients receiving antiretroviral therapy at the treatment centre of the Regional Hospital Bamenda, Northwest Cameroon

**Calvin Bisong Ebai** [1]*, **Cedric Yamssi**[2], **Flore Nguemaïm Ngoufo**[1,2], **Nicaise Ngouemeta Tchoffo**[1], **Omarine Nfor Nlinwe**[1], **Helen Kuokuo Kimbi**[2,3]

1 Department of Medical Laboratory Science, Faculty of Health Sciences, University of Bamenda, Bambili, North West Region, Cameroon, 2 Department of Biomedical Sciences, Faculty of Health Sciences, University of Bamenda, Bambili, North West Region, Cameroon, 3 Department of Animal Biology and Conservation, Faculty of Science, University of Buea, Buea, South West Region, Cameroon

* ebaipi2000@yahoo.com

## Abstract

### Background

Malaria and HIV are leading causes of death in Africa, including Cameroon. Antiretroviral therapy (ART) is expected to boost immunity and reduce vulnerability to opportunistic infections. Reports on comorbidities including malaria are common in Cameroon.

### Objectives

To determine the prevalence of malaria parasitaemia, clinical manifestations, treatment related factors and prevention methods associated with malaria parasitaemia as well as parasite density among HIV patients on ART.

### Methods

It was a cross-sectional study among HIV patients on ART at the Regional Hospital Bamenda. A pre-tested questionnaire was used to collect data on participants' socio-demographic characteristics, clinical manifestations, history of HIV treatment and malaria prevention methods. Microscopy was used for malaria parasite diagnosis and to determine white blood cell (WBC) count. Data was analysed using SPSS version 20.

### Results

The study included 181 participants. The overall prevalence of malaria parasitaemia was 9.4%. Although there were no significant statistical differences, the following trends were observed in the results: 55-71 year old age group (14%) was most infected with malaria parasite with the highest prevalence among those on the 8A first-line ART (10.3%, 14/122); ($\chi 2$=1.03, p=0.96), only participants in the 8A and 12A ART protocols were infected; parasite

**Data Availability Statement:** The dataset is available from Dryad (https://doi.org/10.5061/dryad.d51c5b0b3).

**Funding:** The author(s) received no specific funding for this work.

**Competing interests:** The authors have declared that no competing interests exist.

prevalence was higher in patients with detectable viral load (14.3%, 3/21), (OR=.0.57 (CI: 0.15-2.2), p=0.41), while GMPD varied from 45.7-53.3-±23.1/µl of blood.

## Conclusion

Low malaria parasite prevalence and density were detected amongst HIV patients receiving ART. A systematic malaria test could be helpful to avert morbidity and improve the general health of patients on ART.

## 1. Introduction

Malaria and HIV are two leading causes of death in Africa, including Cameroon [1]. Malaria is a febrile disease caused by parasites of the genus *Plasmodium*. Five species (*Plasmodium falciparum*, *P. malariae*, *P. ovale*, *P. vivax* and *P. knowlesi*) infect humans who present with varied manifestations based on parasite interaction with the infected host's immune system. Human malaria caused by *P. knowlesi*, the most recent species identified in south eastern Asia as a parasite of macaques, is similar to that caused by *P. falciparum* [2]; the most severe form responsible for the majority of deaths related to malaria [3]. It is the most common species in sub-Saharan Africa (SSA) [4], where the greatest burden of malaria is reported [5]. Malaria parasites are transmitted through the bites of an infected female *Anopheles* mosquito during its blood meal. Factors favouring malaria transmission include environmental factors such as the presence of bushes and stagnant ponds of water around residence, warm climate and rainfall, among others. Even so, factors related to human behaviour including farming near homes, urban development with poor drainage, migration, agriculture, dressing that exposes the body to bites of mosquito vectors as well as nocturnal activities (given that the vector is nocturnal, biting from dusk to dawn with peaks at about 10 pm), [6–10] favour malaria parasite transmission. Globally, according to the World Health Organization, in 2022, there were an estimated 249 million malaria cases in 85 malaria-endemic countries and areas; an increase of 5 million cases compared with 2021, causing 608,000 deaths annually [5]. About 94% of malaria cases occurred in the WHO African Region with 508,000 deaths at a rate of 56 per thousand, down from 61 per thousand in 2020 [5]. In Cameroon, an average of 6,459,013 cases of malaria and 12,587 deaths were recorded in 2022 [5].

Bamenda in northwest Cameroon is a hypoendemic region for malaria transmission [11]. Studies have reported varied prevalence of malaria in Bamenda among different groups of individuals. Reports by Ngum *et al*. [12] and Ebai *et al*. [13] have shown 26.7% prevalence among patients consulting at the Regional Hospital Bamenda (RHB) and 18.0% among pregnant women attending antenatal care (ANC) at the same facility, respectively.

The severity of malaria has been associated with the intensity of infected vector bites, and the general health status of the infected person given that parasite load has been found to increase with the number of bites [14]. The parasite, especially *P. falciparum*, is associated with immunosuppression and massive destruction of red blood cells leading to anaemia [15], asthenia, respiratory difficulties and cerebral malaria, in case parasites are sequestrated in deep veins of the brain, among other manifestations [16, 17]. These could be severe sometimes, especially in children less than five years of age, pregnant women, non-immune and immunosuppressed individuals, including those having underlying diseases like HIV/AIDS, in whom there could be more severe disease condition and increased mortality [18]. However, an efficient management of the comorbidities like malaria is key to averting this severity.

Unfortunately, malaria testing is not done as a routine check for HIV patients on treatment in the RHB probably because it is not inscribed in the National Strategic Plan for fight against HIV/AIDS and STIs [18].

HIV/AIDS is an immunosuppressive disease that targets the immune system of the body through the invasion and destruction of CD4+ cells, a clone of T-lymphocytes white blood cells that play an important role in immune responses. CD4+ cells help in the activation of B-lymphocytes to plasma cells, which secrete immunoglobulins responsible for humoral immune responses against infections in the body [19]. HIV transmission is principally through unprotected sexual intercourse, but also through transfusion of infected blood, intravenous (IV) drug use and from mother to child, especially during childbirth, as well as accidental exposure to infected blood [20]. HIV/AIDS patients often present with reduced immunity due to reduction of CD4+ cells over time, thus exposing the patients to opportunistic infections by bacteria, fungi, viruses and parasites [19]. This further leads to severe disease that presents with persistent fever, diarrhoea, weight loss and death, among others [21]. According to the WHO, about 39 million people worldwide live with HIV, 1.3 million new cases were recorded in 2022 with about 630,000 deaths worldwide. Among the people living with HIV (PLHIV) 76.41% (29.8 million) are receiving antiretroviral therapy (ART) [22]. In the WHO African region, an estimated 25.6 million people lived with HIV in 2022, out of which 82% (20.9 million) received ART. Still in the African region, an estimated 660 000 people were infected with HIV in 2022 while 380 000 [300 000–540 000] deaths were attributed to HIV-related causes in 2022 [22].

In Cameroon, an estimated 480 000 people are infected with HIV, 9, 900 new cases of infection were recorded in 2023 with about 2.6% (12,480) deaths [23]. A total of about 312 214 people are on ARVs in Cameroon with about 11% of them in the North West Region where the HIV prevalence is 4% [23]. Some 4000-5000 HIV-positive individuals including about 15 newly diagnosed HIV-positive individuals are received at the treatment centre of the RHB each month where treatment is done using ART as prescribed by the national guidelines. Three therapeutic protocols, divided into first, second and third lines, are available with different combinations of antiretrovirals.

Over the years, the approach to care for HIV patients has improved from treating patients based on reduced levels of CD4+ cells in blood and high viral loads in the late 1990s to operation test and treat all positives (option B+), recently [18]. In the latter option, patients are enrolled to treatment protocols in a treatment centre also known as day hospital once tested and confirmed positive. This approach aims to reduce the morbidity and mortality associated with HIV/AIDS. HIV treatment is expected to boost the immunity of patients by inhibiting virus replication in the body and therefore reduce their vulnerability to other infections. That notwithstanding, there are reports on comorbidities like malaria in HIV patients under treatment in Cameroon with varied prevalence. These include 58.9% in Limbe, Southwest of Cameroon [24], 2.24% in Bamenda, Cameroon in 2012 [25] and 24.5% in Regional Hospital Bamenda in 2018 [26]. Both diseases contribute to immunosuppression with sometimes, exacerbated outcomes [27]. Based on the above, the objectives of this study were to determine the prevalence of malaria parasitaemia, clinical manifestations, treatment related factors and prevention methods associated with malaria parasitaemia as well as the parasite density in HIV patients on ART at the treatment centre of the Regional Hospital Bamenda in northwest of Cameroon. The findings of this study are expected to provide a guide in the conception of an integrated approach to the control of malaria and reduction of morbidity and mortality among HIV/AIDS patients on ART.

## 2. Materials and methods

### 2.1. Study design

This was a cross-sectional study carried out in the HIV treatment centre of the RHB during the months of April and May 2023. The subjects were HIV-positive patients on treatment, enrolled for participation in the study after obtaining their informed consent. A pre-tested questionnaire was used to collect data on socio-demographic characteristics, clinical manifestations, and history of HIV treatment, as well as use of malaria prevention methods.

**2.1.1. Study area and population.** The study was carried out at the HIV/AIDS treatment centre of RHB which is the highest referral hospital in the region and the largest treatment centre in the North West Region of Cameroon. The centre is known to receive on average, 4000-5000 HIV-positive individuals with about 15 newly diagnosed HIV- positive individuals, each month. Bamenda has a predominantly savannah type of vegetation just like the majority of the region and is located at longitude 100.08 to 100.12 E and latitude 50.55 to 60.00 N, at about 1258-1770m above sea level. Bamenda has two seasons, a rainy season that spans from April to November and a dry season from December to March. With an average temperature of 23˚C, March is the warmest month while July is the coldest with an average temperature of 20.1˚C. The average annual rainfall is 2145 mm [28, 29]. Bamenda is the capital of the North West Region of Cameroon, and is a cosmopolitan city with several private and state-owned health facilities and educational establishments. Economic activities are represented at small, medium and large scales, but farming predominates.

The study population was made up of HIV/AIDS-positive individuals of both sexes receiving treatment at the Day Hospital of the RHB who are on one of the following antiretroviral protocols. First-line ART regimens: 7A (Abacavir, lamivudine, Dolukegravir); 8A (Tenofovir, lamivudine, Dolukegravir) and 9A (Tenofovir, lamivudine, Efavirenz). Second-line ART regimens: 10A (Douvir, lamivudine, Dolukegravir); 12A (Douvir, lamivudine, Atazanavir) and third-line ART regimens: 19A (Tenofovir, lamivudine, Dolukegravir). Participants on antimalarial medication within one week to admission in the study were excluded.

**2.1.2. Sampling and sample size estimation.** The convenient sampling technique was used. The sample size was computed using the Cochran formula ($n=Z^2pq/d^2$) [30], where n is the required sample size, Z is 1.96, the standard normal deviation for 95% confidence interval, p =24.5% is the prevalence of malaria among HIV patients on treatment reported by Achare *et al* in a study in Bamenda [26], q=1-p, while d is 0.05 which is the acceptable level of error set for the study. The sample size was calculated at 284 participants.

### 2.2. Data collection

**2.2.1 Administration of questionnaire.** After explaining the study objectives and procedure to eligible participants, a pretested questionnaire was administered to those who consented in order to obtain data on socio-demographic characteristics, clinical manifestations, history of ART, environmental factors and the methods of malaria prevention. The questionnaire was administered in English and exceptionally in Pidgin English and French, for participants who preferred one of these two Languages.

**2.2.2 Laboratory procedures.** Capillary blood sample was collected from each participant by finger prick. 70% alcohol was used to disinfect the tip of the third or fourth finger before pricking with a sterile lancet to obtain blood. Two drops of blood exclusive of the first drop were collected and placed on separate spots on a glass slide, and were used to prepare thick and thin blood smears. The smears were allowed to air dry. After drying, the thin smears were fixed with absolute methanol. Both thick and thin films were stained with 10% Giemsa stain

for 10 minutes. Malaria parasitaemia per µl of blood (MP/ µl) was determined by counting malaria parasite stages against 500 white blood cells and the final value obtained from the formula MP= counted parasites/500*WBC count [31]. Up to 500 WBCs were counted per smear because of the very low parasitaemia observed. Each smear was read by two parasitologists using a binocular light microscope and in case of a discrepancy, the value of a third reader was considered. Another drop of blood was collected to determine the WBC count using the improved Neubauer counting chamber. Briefly, a 1/20 dilution was prepared using a solution of acetic acid tinged in 1% gentian violet. A drop was mounted on the Neubauer counting chamber, WBCs were counted in the four large squares, and WBC count was determined using standard formula [31].

### 2.3. Data analysis

Data was entered into Microsoft Excel Spread Sheet and exported to the Statistical Package for the Social Sciences (SPSS), version 20.0 for analyses. Demographic characteristics such as gender and age were summarized into percentages. The Chi-square test was used to compare proportions while odds ratio (OR) was used to determine risk factors. The variation in geometric mean parasite densities (GMPDs) was determined using one-way Anova test. The level of significance was set at $p < 0.05$.

### 2.4. Ethical considerations

The ethical clearance to carry out this study was obtained from the Institutional Review Board of the University of Bamenda hosted by the Faculty of Health Sciences. An administrative clearance for the study was obtained from the Regional Delegation of Public Health for the North West Region while an authorization was obtained from the RHB administration. Written informed consent was obtained from participants after providing information on the objectives, procedures, risk and advantages of the study. Participation in the study was totally free and voluntary. Participants could quit the study at any time without any penalty. Confidentiality was respected by the use of codes on all documentations. All participants diagnosed of malaria parasitaemia were referred to a clinician for treatment.

## 3. Results

### 3.1. Socio-demographic characteristics of study participants

A total of 181 participants met the criteria for recruitment and took part in this study. The age range was 21-71 years with a mean±SD of 46.53±11. The most represented age group was 21-39(26.5%). There were more females 138(76.2%) than males. Majority of the participants; 158 (87.3%) were resident in Mezam Division of the North West Region. With respect to level of education, 78 (43.1%) of the participants were educated up to primary level. Most of the participants 171 (94.5%) were Christians, no other religion was reported (Table 1).

### 3.2. Prevalence of malaria parasite in HIV positive patients

**3.2.1. Prevalence of malaria parasite in HIV-patients by sociodemographic characteristics.** The overall prevalence of malaria was 9.4% (17/181). Only *Plasmodium falciparum* was detected. Although there was no significant statistical difference in the prevalence of malaria by sociodemographic characteristics, a higher prevalence was observed amongst individuals aged 55-71 years (14%), females (10.1%), those with no formal education (17.6%), persons engaged in business (12.3%) and widows (10.3%) compared to their corresponding counterparts. This is shown in Table 2.

**Table 1. Socio-demographic characteristics of the study participants.**

| Characteristic | Category | Frequency | Percentage |
|---|---|---|---|
| **Statistic** | Mean ±SD | 46.53±11 | |
| | Range | 21-71 | |
| **Age group (years)** | 21-39 | 48 | 26.5 |
| | 40-47 | 47 | 26.0 |
| | 48-54 | 43 | 23.8 |
| | 55-71 | 43 | 23.8 |
| **Sex** | Female | 138 | 76.2 |
| | Male | 43 | 23.8 |
| **Address** | Mezam | 158 | 87.3 |
| | Others | 23 | 12.7 |
| **Level of education** | No formal education | 17 | 9.4 |
| | Primary | 78 | 43.1 |
| | Secondary | 68 | 37.6 |
| | Tertiary | 18 | 9.9 |
| **Religion** | Christianity | 171 | 94.5 |
| | None | 10 | 5.5 |
| **Occupation** | Business | 57 | 31.5 |
| | Farming | 54 | 29.8 |
| | Non salary earning | 48 | 26.5 |
| | Salary earning | 22 | 12.2 |
| **Marital status** | Married | 101 | 55.8 |
| | Single | 51 | 28.2 |
| | Widowed | 29 | 16.0 |
| | Total | 181 | 100.0 |

**3.2.2. Prevalence of malaria parasite by treatment and prevention methods.** Although there was no significant statistical difference in the prevalence of malaria with respect to therapeutic protocols ($\chi^2$=1.03, p=0.96), a higher prevalence was observed in participants who were taking first-line treatment, Tenofovir, Lamivudine, Dolukegravir (8A) (11.5%, 14/122). Participants with malaria parasitaemia were from 8A and 12A ART which are in the first and second lines of ART, respectively. Regarding viral load, the prevalence of malaria was higher in patients with detectable viral load (14.3%, 3/21) than among those with undetectable viral load (8.8%, 14/160), although the difference was as well, not statistically significant (OR=0.57 (CI: 0.15-2.2), p=0.41). With respect to malaria prevention methods reported by the participants, it was observed that participants who live by bush (11.3%, 13/115), those who do not use insecticide spray (10.4%, 16/154) and those who closed doors and windows after 5:00pm (12.9%, 12/110) had a higher malaria prevalence than their respective counterparts, although none of these showed a significant statistical difference as seen in Table 3.

**3.2.3. Prevalence of malaria parasite by clinical manifestations.** Comparing the prevalence of malaria parasitaemia by clinical manifestations, the prevalence was significantly higher statistically, among patients who presented with dyspnoea (25%, 3/12) than those without. Those with dyspnoea were about four times more likely to have malaria parasite than those without (OR= 3.7 (0.9-15.2), p=0.04). There were no significant statistical differences between the prevalence in participants with fever (14.7%, 5/34) who were about two times more likely of having malaria parasite (OR=1.94 (CI: 0.63-5.93), p=0.24); chills (13.3%, 4/30) who were about two times more likely (1.63 (CI: 0.50-5.40), p=0.42) to be infected by malaria parasite; as

**Table 2. Prevalence of malaria parasite by socio-demographic characteristics.**

| Characteristics | Category | No. tested (%) | No. positive (%) | $\chi^2$(p-value) |
|---|---|---|---|---|
| **Age (years)** | 21-39 | 48 (26.5) | 3 (6.3) | 1.99 (0.57) |
| | 40-47 | 47(26.0) | 5 (10.6) | |
| | 48-54 | 43(23.8) | 3 (7.0) | |
| | 55-71 | 43(23.8) | 6 (14.0) | |
| **Sex** | Female | 138(76.2) | 14 (10.1) | 0.38 (0.53) |
| | Male | 43(23.8) | 3 (7.0) | |
| **Address** | Mezam | 158(87.3) | 16 (10.1) | 0.78 (0.37) |
| | Others | 23(12.7) | 1 (4.3) | |
| **Level of education** | No formal education | 17(9.4) | 3 (17.6) | 3.32 (0.34) |
| | Primary | 78(43.1) | 5 (6.4) | |
| | Secondary | 68(37.6) | 6 (8.8) | |
| | Tertiary | 18(9.9) | 3 (16.7) | |
| **Religion** | Christian | 171(94.5) | 16 (9.4) | 0.03 (0.86) |
| | None | 10(5.6) | 1 (10.0) | |
| **Occupation** | Business | 57(31.5) | 7 (12.3) | 3.1 (0.38) |
| | Farming | 54(29.8) | 2 (3.7) | |
| | Non salary earners | 48(26.5) | 5 (10.4) | |
| | Salary earners | 22(12.2) | 3 (13.6) | |
| **Marital status** | Married | 101(55.8) | 9 (8.9) | 0.07 (0.97) |
| | Single | 51(28.2) | 5 (9.8) | |
| | Widowed | 29(16.0) | 3 (10.3) | |
| **Type of house** | Cement | 140 | 14 (10.0) | $\chi^2$ =0.27 (0.61) |
| | Mud brick | 41 | 3 (7.3) | |
| | Total | 181 (100) | 17 (9.4) | |

well as headache (12.3%, 8/65), who were also about 2 times more likely to have malaria parasite (1.70 (CI: 0.61-4.6), p=0.31) than their corresponding counterparts (Table 4).

**3.2.4. Variation in malaria parasite density (GMPD/µl of blood) by viral load, ART protocol and duration.** Comparing the GMPD of participants with respect to treatment-related factors using one-way Anova test showed no significant statistical differences. However, there was a general tendency for GMPD to increase with duration of treatment. Patients who had been on treatment for at least 4 years had the highest GMPD (50±20 parasites/µl). Only patients on the 8A and 12A therapeutic protocols were infected with malaria parasite with GMPD of 48.6±17.5 and 40±00 per µl of blood, respectively. There was also a tendency for higher GMPD among participants with detectable viral load (53.3±23.1 parasites/µl of blood) than those with undetectable viral load (45.7±14.5 parasites /µl of blood) (Table 5).

## 4. Discussion

Malaria and HIV/AIDS co-infections continue to occur despite the measures put in place by National and sub-national governments to fight against these diseases which are endemic in sub-Saharan Africa including Cameroon. Nonetheless, these measures are yielding remarkable results as can be seen with the relatively low malaria parasite prevalence and densities recorded in this study. An overall malaria parasitaemia prevalence of 9.4% was observed in this study; lower than the 58.9% reported by Kimbi *et al.* over a decade ago [24] and the 24.5% by Achere *et al.*, recently [26], both among HIV/AIDS patients on ART in Cameroon. There has been a progressive decrease in malaria prevalence on the global and national scales due to a scale-up

**Table 3. Prevalence of malaria parasite by ART protocol, viral load and malaria prevention methods.**

| Factor/method | Category | N°. tested | No. positive (%) | Test | P-value |
|---|---|---|---|---|---|
| **ARV** | 7A | 1 | 0 (0.0) | $\chi^2$=1.03 | 0.96 |
| | 8A | 122 | 14 (11.5) | | |
| | 9A | 3 | 0 (0.0) | | |
| | 10A | 1 | 0 (0.0) | | |
| | 12A | 34 | 3 (8.8) | | |
| | 19A | 3 | 0 (0.0) | | |
| **Last viral load** | Detectable | 21 | 3 (14.3) | OR=.0.57 0.15-2.2) | 0.41 |
| | Undetectable | 160 | 14 (8.8) | | |
| **Live by bush** | No | 66 | 4 (6.1) | $\chi^2$=1.36 | 0.24 |
| | Yes | 115 | 13 (11.3) | | |
| **Live by stagnant water** | No | 145 | 14 (9.7) | $\chi^2$=0.59 | 0.89 |
| | Yes | 36 | 3 (8.3) | | |
| **Use mosquito net** | No | 68 | 7 (10.3) | $\chi^2$=0.001 | 0.98 |
| | Yes | 96 | 10 (10.4) | | |
| **Use insecticide spray** | No | 154 | 16 (10.4) | $\chi^2$=1.21 | 0.27 |
| | Yes | 27 | 1 (3.7) | | |
| **Close doors and windows** | After 5pm | 110 | 12(10.9) | $\chi^2$=0.06 | 0.81 |
| | Before 5pm | 71 | 5(7.0) | | |

First-line ART: 7A=Abacavir, lamivudine, Dolukegravir; 8A=Tenofovir, lamivudine, Dolukegravir; 9A= Tenofovir, lamivudine, Efavirenz. Second-line ART: 10A=Douvir, lamivudine, Dolukegravir; 12A=Douvir, lamivudine, Atazanavir. Third-line ART: 19A= Tenofovir, lamivudine, Dolukegravir.

of the measures to combat the disease [23]. Moreover, in the past, ART was not free while, there has been a lot of education through mass media, especially in hospital settings. This might have benefitted the masses. However, an upsurge in prevalence was observed in 2020

**Table 4. Prevalence of malaria parasite by clinical manifestations.**

| Clinical manifestation | Category | No. tested | No. Positive (%) | OR (95% CI) | p-value |
|---|---|---|---|---|---|
| **Fever** | No | 147 | 12 (8.2) | 1.94 (0.63-5.93) | 0.24 |
| | Yes | 34 | 5 (14.7) | | |
| **Chills** | No | 151 | 13 (8.6) | 1.63 (0.50-5.40) | 0.42 |
| | Yes | 30 | 4 (13.3) | | |
| **Abdominal pain** | No | 151 | 14 (9.3) | 1.10 (0.29-4.04) | 0.90 |
| | Yes | 30 | 3 (10.0) | | |
| **Headache** | No | 116 | 9 (7.80) | 1.70 (0.61-4.6) | 0.31 |
| | Yes | 65 | 8 (12.3) | | |
| **Nausea** | No | 167 | 14 (8.4) | 3.0 (0.74 -11.95) | 0.11 |
| | Yes | 14 | 3 (21.4) | | |
| **Diarrhoea** | No | 157 | 14 (8.9) | 1.50 (0.39 -5.50) | 0.57 |
| | Yes | 24 | 3 (12.5) | | |
| **Dyspnoea** | No | 169 | 14 (8.3) | 3.70 (0.90 – 15.2) | 0.04 |
| | Yes | 12 | 3 (25.0) | | |
| **Fatigue** | No | 113 | 8 (7.1) | 2.0 (0.73 – 5.50) | 0.17 |
| | Yes | 68 | 9 (13.2) | | |
| **Joint/muscle pain** | No | 134 | 10 (7.5) | 2.17 (0.77 – 6.07) | 0.13 |
| | Yes | 47 | 7 (14.9) | | |

**Table 5. Variation in GMPD by viral load, ART protocol and duration.**

| Factor | Category (n) | GMPD (±SD) | F | p-value |
|---|---|---|---|---|
| Duration of treatment (years) | 1(2) | 40.0 (0.0)) | 0.3 | 0.59 |
| | 2(5) | 48.0 (17.9) | | |
| | 3 (6) | 46.7 (16.3) | | |
| | 4 (4) | 50.0 (20.0) | | |
| Treatment type | 8A (14) | 48.6 (17.5) | 0.72 | 0.41 |
| | 12A (3) | 40.0 (0.0) | | |
| Viral load | Detectable (3) | 53.3 (23.1) | 0.56 | 0.46 |
| | Undetectable (14) | 45.7 (14.5) | | |

due to a breakdown in health systems as a result of the Covid 19 pandemic [5]. The prevalence of malaria parasitaemia in this study was higher compared to the 2.2% reported by Njunda *et al.* in 2012 among PLWHIV in Bamenda [25]. This finding is surprising and cannot be explained. The prevalence of malaria parasitaemia in this study was also higher than the 7.8% reported by Akinbo *et al.* in 2016 in Kogi state, Nigeria [32]. The higher prevalence observed in the present study could be due to variability in the dynamics of malaria transmission in different ecological settings, as well as variations in the strengths of different national health systems.

Regarding socio-demographic characteristics, the absence of significant variation in prevalence of malaria among the participants in this study is similar to findings from other studies. In similar previous studies by Sahle *et al.* in Ethiopia [33] and Simon-Oke *et al.* in Nigeria [34] there was no significant statistical difference between age groups among participants in similar studies. Kwenti *et al.* [35] in a study in Yaounde, Cameroon, and Gumel *et al.* in northwest Nigeria [36] reported no significant statistical difference in malaria parasite prevalence between gender and occupation, respectively. There is a likelihood that the ART received, placed participants in similar states in terms of defence against infections (malaria in the present study). With respect to address of residence, there was no significant statistical difference between participants who live in Mezam Division and other Divisions. This could be due to the fact that a greater expanse of the Region is endowed with similar environmental conditions. Furthermore, there could be a general sense of awareness by the patients of their precarious state of the health that compels them to adopt similar healthy behaviours. Given that ART is prescribed following individual particularities, socio-demographics may not play a significant role on the vulnerability of the patient to malaria. However, the study by Akinbo *et al.* in Nigeria [32], reported a significant statistical difference in malaria prevalence with respect to gender.

This study showed that there were no significant statistical differences in malaria prevalence and parasite densities (GMPD) between ART protocols as well as viral loads. There is a possibility of similarity in immune status enhancement afforded by ART to patients and the malaria prevention methods adopted by the patients. Similar findings were reported by Tchinda *et al.* in Douala, Cameroon [37], Jegede *et al.* in Nigeria [38] and Ejigu *et al.* in southwest Ethiopia [39]. Detection of malaria parasitaemia among patients on 8A and 12A ART protocols may have been due to the fact that the majority of the study participants were enrolled into these two. Lower prevalence and density of parasitaemia among patients with undetectable viral load could be an indication of improved immunity.

There was a tendency for malaria parasite prevalence to be lower among those who closed their doors and windows before 5pm, those who used bed nets, and those who used insecticidal sprays. These measures are evidently protective against malaria transmission. Studies

have shown varied reports on this; Jegede *et al.* [38], showed no significant statistical difference while Akinbo *et al.* [32] reported significant statistical difference. This can be due to peculiarities in study populations and other factors including level of endemicity of malaria in the different areas.

The higher prevalence of malaria parasitaemia observed among participants with dyspnoea is an indication that malaria is associated with difficulty in respiration. This could be due to an increased respiratory rate associated with low levels of haemoglobin, which is common in *Plasmodium falciparum* infection as a result of massive destruction of red blood cells [33]. It is pertinent to state that the other clinical manifestations for which there were no statistical differences in malaria parasite prevalence, may have been due to the HIV infection, and also the type of activity the participants engaged in, given that most of them were into business, farming and non-salary earnings which are generally associated with some physical activity that may contribute to fatigue, pains, headache and other manifestations.

It should be noted that, no confirmation of malaria parasitaemia using PCR was done apart from microscopy. This might have limited the chances of detecting parasitaemia in patients with lower malaria parasite densities. Also, data on clinical manifestations were based on participants' responses which being subjective may have affected the results. The sample size was not attained due to patients' refusal to participate.

## 5. Conclusion

Results of the present study show that there were low levels of malaria parasite prevalence and density amongst HIV patients on ART in the treatment centre of the Bamenda Regional Hospital. Apart from dyspnoea as a clinical manifestation, the prevalence of malaria parasitaemia and parasite density were similar among patients, although tendencies of variation in manifestations were observed between some groups. A systematic malaria test could be helpful in averting morbidity and improving on the general health of patients on ART.

## Acknowledgments

The authors acknowledge the authorities of the Regional Hospital Bamenda, for accepting that this study should be carried out in the facility. We also acknowledge the participants for accepting to take part in the study.

## Author Contributions

**Conceptualization:** Calvin Bisong Ebai.

**Data curation:** Calvin Bisong Ebai, Nicaise Ngouemeta Tchoffo.

**Formal analysis:** Calvin Bisong Ebai.

**Investigation:** Calvin Bisong Ebai.

**Methodology:** Calvin Bisong Ebai, Nicaise Ngouemeta Tchoffo.

**Project administration:** Calvin Bisong Ebai.

**Resources:** Calvin Bisong Ebai, Nicaise Ngouemeta Tchoffo.

**Software:** Calvin Bisong Ebai.

**Supervision:** Calvin Bisong Ebai, Helen Kuokuo Kimbi.

**Validation:** Calvin Bisong Ebai.

**Visualization:** Calvin Bisong Ebai.

**Writing – original draft:** Calvin Bisong Ebai.

**Writing – review & editing:** Calvin Bisong Ebai, Cedric Yamssi, Flore Nguemaïm Ngoufo, Nicaise Ngouemeta Tchoffo, Omarine Nfor Nlinwe, Helen Kuokuo Kimbi.

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
