## [Decision Letter · Decision Letter 0]

8 Apr 2024

PONE-D-24-04976Low levels of Plasmodium falciparum parasitemia among patients receiving antiretroviral therapy at the treatment center of the Regional Hospital Bamenda, Northwest CameroonPLOS ONE

Dear Dr. CALVIN EBAI,

Thank you for submitting your manuscript to PLOS ONE. After careful consideration, we feel that it has merit, but in its present form, the manuscript is not suitable for publication in PLOS ONE until after a major revision addressing critical comments raised by the reviewers. Therefore, we invite you to submit a revised version of the manuscript that addresses the points raised during the review process.

We look forward to receiving your revised manuscript.

Kind regards,

Segun Isaac OYEDEJI, Ph.D

Academic Editor

PLOS ONE

Journal Requirements:

2. Please amend either the abstract on the online submission form (via Edit Submission) or the abstract in the manuscript so that they are identical.

**Additional Editor Comments**:

In addition to the Reviewers' comments outlined below, please attend to the following:

ABSTRACT

Page 1: Lines 24 and 25

Please replace the sentence "Capillary blood was used for malaria parasite diagnosis and to determine white blood cell count" with the actual method that was used for malaria parasite diagnosis, which in this case, is microscopy.

Page 2: Lines 35 and 36

Please remove the sentence "The prevalence of malaria parasitaemia and parasite density was similar despite variations among the patients." from conclusion. You may wish to incorporate it in the Results section.

INTRODUCTION

Page 2: Line 41

The reference [1], cited in support of the statement is inadequate. Please verify.

Page 3: Line 74

Please replace "susceptible" with another term, such as akin, tantamount, etc., to appropriately convey the message.

Page 4: Lines 79 and 80

Please delete the parenthesis (white blood cells). At this level, that knowledge is implied.

RESULTS

In TABLE 1 (Page 10), and Page 9: Line 204, why did the authors provide only two categories for religion: Christianity and None? What about the other forms of religion? I will advise that, if possible, this section of the data should be removed, because only one type of religion was involved and could be regarded as biased.

DISCUSSION

Page 17: Line 272

Please replace "these measures" with "the measures" since they are not listed contextually.

**Reviewers' comments**:

Reviewer's Responses to Questions

**Comments to the Author**

1. Is the manuscript technically sound, and do the data support the conclusions?

Reviewer #1: Yes

Reviewer #2: Partly

2. Has the statistical analysis been performed appropriately and rigorously? 

Reviewer #1: Yes

Reviewer #2: No

3. Have the authors made all data underlying the findings in their manuscript fully available?

Reviewer #1: Yes

Reviewer #2: No

4. Is the manuscript presented in an intelligible fashion and written in standard English?

Reviewer #1: Yes

Reviewer #2: No

5. Review Comments to the Author

Reviewer #1: Introduction:

L53 -56: Authors should add migrant and agricultural workers, these are two factors that also contribute to transmission of malaria.

L67: “The severity …..associated with infected vector bites” What do the authors mean by this?

L70: Cerebral malaria is usually the resultant effect of parasite sequestration in into the deep veins and their ability to cross the BBB. Authors should please correct.

Results:

Prevalence of malaria parasites in HIV positive patients. I am still trying to understand why despite the huge difference between malaria prevalence in Female V Male, Mezeam V Others, Christian V None, there is no statistical significance. The authors should please check these calculations again.

Discussion

The discussion should be structured in the way that the results were presented. First, following the general overview of the demographics of the study participants, the authors should discuss how the prevalence of malaria varies in each group: Females/Males, Mezeam/Other residents, Christian/Others etc. And also contextualised the reasons for the observed differences, for instance, is there any biological or behavioural reasons responsible for the observed prevalence in females and males, this should be done also with others.

For future studies, the authors should incorporate PCR method of malaria testing as Cameroon is also known for the transmission of non-falciparum malaria.

Is there any contra-indication of ACTs with any f the drugs used for treating PLHIV? If not, the authors should also add to their recommendation the need to test and treat PLHIV for malaria when they visit hospitals.

Reviewer #2: The authors are referred to the attached reviewed manuscript (please see track changes) for my detailed comments on all sections of the manuscript. However, please find below, a few of the specific comments/concerns and suggestions:

• The conceptualization of the study was adequate but the execution was inadequate. The manuscript needs to be thoroughly revised to make for a better article. Please see attached "Reviewed" manuscript.

• The authors’ recourse to spelling of the word ‘’Parasitemia’’ interchangeably with ‘’Parasitaemia’’ in the manuscript is not ideal. For consistency and depending on Journal style, the authors should adopt only one spelling; American (parasitemia) or British (parasitaemia).

• Line 41: Reference [1] is exclusive to the assertion on HIV as a leading cause of mortality. The authors need to back up a similar assertion on malaria with a reference.

• Line 61: The authors should double-check the highlighted figures/estimates (6.459.013 and 12.587) for correctness. Could be typos.

• Lines 74/75: The highlighted sentence is vague and needs to be recast for better comprehension.

• Line 90: The authors should please reconcile these estimates 89% (29.8 million).

• The objectives of the study were clearly stated and the study design described. However, the clinical manifestations and factors associated with malaria parasitaemia among HIV patients on ART, as described in the study, were not appropriate/adequate to address the stated objectives. Moreover,

• Line 168/169: Authors need to be more specific. They did not state whether or not the two drops of blood for preparation of blood smears were inclusive/exclusive of the very first drop of blood after pricking the finger with a sterile lancet?

• Line 171: ‘’Both thick and thin films were stained with 10% Giemsa stain …..’’

• Although this yields faster results, the films may be of less predictable quality than if stained with 2.5% or 3% Giemsa stain. One would have expected that for such a study, either of the latter concentrations would have been used.

• Actual sample size (181) was far less than the estimated sample size (284). This was a limitation which was beyond the authors, as the number of participants would depend on the eligibility criteria and voluntariness of individuals to be part of the study. This limitation was stated by the authors in the results and conclusion. Some other valid limitations which could affect the outcome of the study were stated in the last paragraph of "Discussion" section.

• The discussion was fraught with some vague statements and assertions not backed up by correct statistical analyses and references (Please see track changes in reviewed manuscript).

• Some of the figures (i.e., %) given in Tables 1-3 are wrong (Please see track changes in reviewed manuscript for corrections).

• Some revisions to the Tables have also been suggested.

6. PLOS authors have the option to publish the peer review history of their article (what does this mean?). If published, this will include your full peer review and any attached files.

Reviewer #1: No

Reviewer #2: No

---

## [Author Response · Author response to Decision Letter 0]

12 Jul 2024

5. Review Comments to the Author

Reviewer #1: Introduction:

L53 -56: Authors should add migrant and agricultural workers, these are two factors that also contribute to transmission of malaria.

L67: “The severity …..associated with infected vector bites” What do the authors mean by this?

L70: Cerebral malaria is usually the resultant effect of parasite sequestration in into the deep veins and their ability to cross the BBB. Authors should please correct.

Results:

Prevalence of malaria parasites in HIV positive patients. I am still trying to understand why despite the huge difference between malaria prevalence in Female V Male, Mezeam V Others, Christian V None, there is no statistical significance. The authors should please check these calculations again. –Verified ok. None of the results showed a statistically significant difference

Discussion

The discussion should be structured in the way that the results were presented. First, following the general overview of the demographics of the study participants, the authors should discuss how the prevalence of malaria varies in each group: Females/Males, Mezeam/Other residents, Christian/Others etc. And also contextualised the reasons for the observed differences, for instance, is there any biological or behavioural reasons responsible for the observed prevalence in females and males, this should be done also with others.

Very correct! Actually, you can see that the discussion is structured in same way as results. We started with the discussion of the overall results then with respect to socio-demographice characteristics. However, among all the socio-demographic characteristics, none showed significant statistical difference, so we chose to discuss some including age and sex based on the trends observed. 

For future studies, the authors should incorporate PCR method of malaria testing as Cameroon is also known for the transmission of non-falciparum malaria.

Agreed

Is there any contra-indication of ACTs with any f the drugs used for treating PLHIV? If not, the authors should also add to their recommendation the need to test and treat PLHIV for malaria when they visit hospitals.

This is important. Although there are reports on the safety of ACT in patients on ARTs, we cannot claim to be exhaustive in our literature search on this issue. So, we decided not to recommend test and treat for PLHIV.

Reviewer #2: The authors are referred to the attached reviewed manuscript (please see track changes) for my detailed comments on all sections of the manuscript. However, please find below, a few of the specific comments/concerns and suggestions:

• The conceptualization of the study was adequate but the execution was inadequate. The manuscript needs to be thoroughly revised to make for a better article. Please see attached "Reviewed" manuscript.

• The authors’ recourse to spelling of the word ‘’Parasitemia’’ interchangeably with ‘’Parasitaemia’’ in the manuscript is not ideal. For consistency and depending on Journal style, the authors should adopt only one spelling; American (parasitemia) or British (parasitaemia).

Noted. That has been corrected.

• Line 41: Reference [1] is exclusive to the assertion on HIV as a leading cause of mortality. The authors need to back up a similar assertion on malaria with a reference.

True. The appropriate reference has been inserted

• Line 61: The authors should double-check the highlighted figures/estimates (6.459.013 and 12.587) for correctness. Could be typos.

Those were typos

• Lines 74/75: The highlighted sentence is vague and needs to be recast for better comprehension.

The word “susceptible” has been replaced by “akin” for comprehension

• Line 90: The authors should please reconcile these estimates 89% (29.8 million).

Agreed! Corrected

• The objectives of the study were clearly stated and the study design described. However, the clinical manifestations and factors associated with malaria parasitaemia among HIV patients on ART, as described in the study, were not appropriate/adequate to address the stated objectives. 

The objectives have been reviewed to suit the factors that were measured. “factors associated” has been replaced with “treatment related factors and prevention measures”

Moreover,

• Line 168/169: Authors need to be more specific. They did not state whether or not the two drops of blood for preparation of blood smears were inclusive/exclusive of the very first drop of blood after pricking the finger with a sterile lancet?

The two drops of blood were exclusive of the first drop after pricking the finger with a sterile lancet.

• Line 171: ‘’Both thick and thin films were stained with 10% Giemsa stain …..’’

Reviewed.

• Although this yields faster results, the films may be of less predictable quality than if stained with 2.5% or 3% Giemsa stain. One would have expected that for such a study, either of the latter concentrations would have been used.

Agreed. This is considered a limit of this study.

• Actual sample size (181) was far less than the estimated sample size (284). This was a limitation which was beyond the authors, as the number of participants would depend on the eligibility criteria and voluntariness of individuals to be part of the study. This limitation was stated by the authors in the results and conclusion. Some other valid limitations which could affect the outcome of the study were stated in the last paragraph of "Discussion" section.

The limitations are presented in the last paragraph of the “Discussion section”

• The discussion was fraught with some vague statements and assertions not backed up by correct statistical analyses and references (Please see track changes in reviewed manuscript).

• Some of the figures (i.e., %) given in Tables 1-3 are wrong (Please see track changes in reviewed manuscript for corrections).

• Some revisions to the Tables have also been suggested.

All figures in table 1 have been verified, all are correct.

Figures on tables 2 and 3 have been verified and corrected. There were typos so statistical test values were not affected by the corrections. All other tables were verified. 

6. PLOS authors have the option to publish the peer review history of their article (what does this mean?). If published, this will include your full peer review and any attached files.

Do you want your identity to be public for this peer review? For information about this choice, including consent withdrawal, please see our Privacy Policy.

Reviewer #1: No

Reviewer #2: No

---

## [Decision Letter · Decision Letter 1]

29 Oct 2024

PONE-D-24-04976R1Low levels of Plasmodium falciparum parasitemia among patients receiving antiretroviral therapy at the treatment center of the Regional Hospital Bamenda, Northwest CameroonPLOS ONE

Dear Dr. CALVIN EBAI,

Once again, thank you for submitting your manuscript to PLOS ONE. The manuscript has undergone a second Review and most of the initial comments have been addresed. However, there are some aspects that still need minor correcttions as highlighted below by one of the Reviewers. Kindly address these aspectsas soon as possible..We invite you to submit a revised version of the manuscript that addresses the points raised during the review process.

We look forward to receiving your revised manuscript.

Kind regards,

Segun Isaac OYEDEJI, Ph.D

Academic Editor

PLOS ONE

Journal Requirements:

Reviewers' comments:

Reviewer's Responses to Questions

**Comments to the Author**

1. If the authors have adequately addressed your comments raised in a previous round of review and you feel that this manuscript is now acceptable for publication, you may indicate that here to bypass the “Comments to the Author” section, enter your conflict of interest statement in the “Confidential to Editor” section, and submit your "Accept" recommendation.

Reviewer #1: All comments have been addressed

Reviewer #2: (No Response)

2. Is the manuscript technically sound, and do the data support the conclusions?

Reviewer #1: Yes

Reviewer #2: Partly

3. Has the statistical analysis been performed appropriately and rigorously? 

Reviewer #1: Yes

Reviewer #2: Yes

4. Have the authors made all data underlying the findings in their manuscript fully available?

Reviewer #1: Yes

Reviewer #2: No

5. Is the manuscript presented in an intelligible fashion and written in standard English?

Reviewer #1: Yes

Reviewer #2: No

6. Review Comments to the Author

Reviewer #1: (No Response)

Reviewer #2: Please find below, detailed comments on the clean copy of the revised manuscript.

General:

• The English language quality (including punctuations, etc.) just average and needs thorough editing and improvement. There is still a need for consistency in spellings throughout the manuscript; examples include dyspnea/dyspnoea; diarrhea/diarrhoea; parasitemia/parasitaemia at different points in the manuscript. These were pointed out in the earlier review.

• There is a mix-up of in-text citations and corresponding references, some of which are listed below. Hence, there is a need for a painstaking audit and reconciliation of all in-text citations and corresponding references to ensure correctness.

Abstract:

Line 15: ART … is undefined. It would be necessary to have this acronym preceded by its full meaning while ART is placed in parentheses, i.e., “Antiretroviral therapy (ART)”. The acronym suffices, subsequently.

Introduction:

• There is a need to underpin the statement in lines 67-70: “The parasite, especially P. falciparum, is associated with immunosuppression and massive destruction of red blood cells leading to anaemia, asthenia, respiratory difficulties and cerebral malaria, in case parasites are sequestrated in deep veins of the brain, among other manifestations.” with relevant citation(s)/reference(s) as suggested in the earlier review.

• Line 72: Replace “for which” with “in whom”

• Lines 73-74: Recast as “However, an efficient management of the comorbidities is key to averting this severity.”

• Lines 75-77: The statement preceding the in-text citation [15] does not align with the citation. The statement is underpinned more by [14] (please refer to lines 379-380 under References). The authors need to reconcile this.

• Lines 81-83: The statement “HIV transmission is principally through unprotected sexual intercourse, but also through transfusion of infected blood, intravenous (IV) drug use and from mother to child, especially during childbirth, as well as accidental exposure to infected blood [17].”) does not align with the citation [17]. The publication [i.e., 17] was on CD4+ and nowhere in the article was reference made to HIV transmission.

• Lines 94-95: The statement “In Cameroon, an estimated 480 000 people are infected with HIV, 9, 900 new cases of infection were recorded in 2023 with about 2.6% (12,480) deaths [21].” The citation may need to be reconciled with both [21] and [22] under References for correctness.

• Line 197: [15] is a wrong citation for the preceding statement which aligns more with [14] under References (please refer to lines 379-380).

• Line 104: [15] is a wrong citation for the preceding statement which is on HIV. [15] is a study on malaria!

• Lines 110-111: [19], [20], and [21] are wrong citations. The sentence in these lines align more with [23], [24] and [25], respectively. Please double-check for correctness.

Materials and Methods

Line 135: The in-text citations [26,27] do not align with the preceding statement but more with [27,28] (please refer to References). Please reconcile/double-check.

Lines 147-148: [25] does not align with Cochran formula which can be found in [29] under References.

Line 150: [21] does not align with any study in Bamenda by Eyong et al. This citation, i.e., [21] is a World Health Organization publication.

Lines 154-157: Recast the sentence to “After explaining the study objectives and procedures to eligible participants, a pretested questionnaire was administered to those who consented in order to obtain data on socio-demographic characteristics, clinical manifestations, history of ART, environmental factors and the methods of malaria prevention.”

Line 162: … to disinfect the tip of the finger before pricking. Which of the fingers? The authors need to state this.

Lines 168 and 175: [29] does not align with the preceding statement. This citation, i.e., [29] is “Cochran WG. Sampling Techniques, John Wiley and Sons Inc., New York; 1963.”

Line 171: delete “)”

Results

Table 1: Under religion, None, the 5.6 should be 5.5. Please note that 10/181 = 5.52486. This was pointed out in the earlier review.

Table 2: Under χ2 (p-value) for Type of house, amend χ2 =0.27 to 0.27 (0.61).

Discussion

Lines 265-266: [19] and [21] do not align with Kimbi et al. and Achere et al., respectively, as stated. They are both WHO publications, while Kimbi et al. and Achere et al. are placed as [23] and [25], respectively (please refer to References).

Lines 268-269: Please recast to “combat the disease [31]. Moreover, in the past, ART was not free while there has been a lot of education through the mass media, especially in hospital settings. This might have benefited the masses”.

Line 271: Please recast to “2020 due to a breakdown in health systems as a result of the Covid 19 pandemic [???].”

Line 271: [17] does not align with the preceding statement.

Line 273: [20] is not Njunda et al. in 2012, in References. It is actually [24].

Line 275: [30] is not Akinbo et al. in 2016, in References. It is actually [31].

Line 280: [33] and [34] are not Sahle et al. and Simon-Oke et al. respectively, in References. They are actually [34] and [35], respectively.

Ditto, all in-text citations and corresponding references onward. The authors need to do a comprehensive review of the in-text citations so as to correspond with their listing in References.

Lines 305-307: Please recast to: “Studies have shown varied reports on this; Jegede et al. [39] showed no significant statistical difference while Akinbo et al. [31] reported significant statistical difference.”

Line 316: Please amend as follows: “and non-salary earnings which are generally associated with some physical activity that may contribute”

References

The correct citations are as follows:

1. Global Health Estimates 2020: Deaths by Cause, Age, Sex, by Country and by Region, 2000-2019. Geneva, World Health Organization; 2020.

37. Gumel SD, Ibrahim A, Olayinka AT, Ibrahim MS, Balogun MS, Dahiru A, et al. HIV-malaria 450 co-infection and its determinants among patients attending antiretroviral treatment 451 clinic in Zaria, Kaduna State, Nigeria. Journal of Interventional Epidemiology and Public Health, 2021; 4: 2. DOI: https://doi.org/10.37432/jieph.2021.4.1.31.

A click at the above DOI shows the publication titled as above. The DOI link does not correspond to the citation provided by the authors. The authors may wish to access the link for correctness.

38. Tchinda GG, Atashili J, Achidi EA, Kamga HL, Njunda AL, Ndumbe PM. Impact of Malaria on Hematological Parameters in People Living with HIV/AIDS Attending the Laquintinie Hospital in Douala, Cameroon. PLoS ONE, 2012; 7(7): e40553. https://doi.org/10.1371/journal.pone.0040553.

A visit to the link provided in the citation by the authors (i.e., Available from: https://www.ncbi.nlm.nih.gov/pmc/articles/PMC3393653/https://doi.org/10.1371/journal.pone.0040553)., shows that the proper citation is as stated above.

• Line 392: Amend manifestation to manifestations.

• Line 394: Amend https://doi.org/10.172226/938. to https://doi.org/10.17226/938.

• Line 421: the correct citation is: Sci Rep. 2021, 11: 16655. DOI:10.1038/s41598-021-95591-6.

• Lines 446-447: the correct citation is …. 44: 156. DOI:10.1186/s42269-020-00401-0.

• Lines 464 and 468: Amend PLOS to PLoS

7. PLOS authors have the option to publish the peer review history of their article (what does this mean?). If published, this will include your full peer review and any attached files.

Reviewer #1: No

Reviewer #2: No

---

## [Author Response · Author response to Decision Letter 1]

20 Nov 2024

All comments by editors and reviewer have been addressed. The comments relating to that are submitted to the attachment labelled Response to reviewers

---

## [Editor Report · Decision Letter 2]

22 Dec 2024

Low levels of Plasmodium falciparum parasitemia among patients receiving antiretroviral therapy at the treatment center of the Regional Hospital Bamenda, Northwest Cameroon

PONE-D-24-04976R2

Dear Dr. CALVIN EBAI,

We’re pleased to inform you that your manuscript has been judged scientifically suitable for publication and will be formally accepted for publication once it meets all outstanding technical requirements.

Kind regards,

Segun Isaac OYEDEJI, Ph.D

Academic Editor

PLOS ONE

---

## [Editor Report · Acceptance letter]

17 Jan 2025

PONE-D-24-04976R2 

PLOS ONE

Dear Dr. CALVIN EBAI, 

I'm pleased to inform you that your manuscript has been deemed suitable for publication in PLOS ONE. Congratulations! Your manuscript is now being handed over to our production team.

Kind regards, 

on behalf of

Professor Segun Isaac OYEDEJI 

Academic Editor

PLOS ONE